# The Impact of Work-Related Problems on Burnout Syndrome and Job Satisfaction Levels among Emergency Department Staff

**DOI:** 10.3390/bs13070575

**Published:** 2023-07-11

**Authors:** Cosmina-Alina Moscu, Virginia Marina, Aurelian-Dumitrache Anghele, Mihaela Anghele, Liliana Dragomir, Anamaria Ciubară

**Affiliations:** 1Emergency Department of Hospital, “Dunărea de Jos” University of Galati, 47 Str. Domnească, 800201 Galati, Romania; cosmina_caluian@yahoo.com; 2Medical Department of Occupational Health, Faculty of Medicine and Pharmacy, “Dunărea de Jos” University of Galati, 47 Str. Domnească Galati, 800201 Galati, Romania; 3Department of General Surgery, Faculty of Medicine and Pharmacy, “Dunărea de Jos” University, 800201 Galati, Romania; anghele_aurelian@yahoo.com; 4Clinical-Medical Department, Faculty of Medicine and Pharmacy, “Dunărea de Jos” University of Galati, 800201 Galati, Romania; mihaela.anghele@ugal.ro (M.A.); lilianadragomir2017@gmail.com (L.D.); anamburlea@yahoo.com (A.C.)

**Keywords:** burnout syndrome, professional satisfaction, healthcare workers, emergency department

## Abstract

Burnout syndrome is caused by a number of factors, including personal, organizational, and professional problems. The purpose of this study was to investigate burnout and satisfaction levels among emergency department staff in the context of professional activity issues. We studied a sample of 184 participants working in the emergency department. Participants signed an informed consent form, completed a socio-demographic questionnaire and the MBI-HSS (MP) questionnaire to assess burnout, a JSS (Job Satisfaction Survey) to assess their professional satisfaction, and the AWS (AWS of work survey) questionnaire on work aspects. The questionnaires were completed between November 2022 and March 2023. The study group was aged between 24–64 years old, most of the subjects being female. The study found that 30.2% of emergency medical staff are at high risk of burnout. Emotional exhaustion is indirectly proportional to workload, interpersonal relationships, and rewards. Emotional exhaustion is a direct result of work experience. While participants expressed ambivalence and dissatisfaction with the work environment, they were satisfied with the nature of their work. This study found that job-related factors such as social support and feedback are significant predictors of employee well-being and reducing the risk of burnout. Emotional exhaustion was negatively correlated with job satisfaction, while personal accomplishment was positively correlated with job satisfaction. Depersonalization was associated with job dissatisfaction with operating conditions and coworkers. The study also identified differences in burnout and related constructs among healthcare professionals, with resident physicians reporting higher levels of personal accomplishments and paramedics reporting relatively low levels of emotional exhaustion. The findings suggest that tailored interventions addressing job demands and resources are critical in improving employee well-being and reducing burnout.

## 1. Introduction

Burnout syndrome has become a pressing concern in many professions, particularly in the healthcare industry, where the demands and stress of the workplace can affect employee well-being. Emergency department staff are particularly vulnerable to burnout due to the high-pressure nature of their work [1,2]. 

Research has shown that burnout is a complex phenomenon that can arise from various factors, including personal, organizational, and professional issues [3]. In the healthcare industry, burnout has been identified as a significant problem among healthcare providers, particularly those working in high-stress environments such as emergency departments [1,2]. To better understand the factors contributing to burnout among emergency department staff, a recent study investigated the impact of work-related problems on burnout and satisfaction levels. The study’s findings shed light on the challenges faced by emergency department staff and highlight the need for healthcare organizations to address these issues to improve the well-being of their staff [4].

Burnout and job satisfaction are two critical factors that influence the well-being and performance of employees in any organization. While burnout has been associated with various adverse outcomes, such as decreased productivity and job satisfaction, research has also shown that job satisfaction can serve as a protective factor against burnout [5]. Understanding the relationship between burnout and job satisfaction is crucial for organizations to create a positive work environment and ensure the well-being of their employees. In this context, a recent study aimed to evaluate the relationship between burnout syndrome and job satisfaction in the context of the domains of working life [6,7].

Work–life balance is essential in the modern workplace, and research has highlighted its impact on employee well-being and job satisfaction [8]. However, work-life balance is only one aspect of an employee’s work-life. This study explored how different aspects of work-life, such as workload, work demands, and social support, are related to burnout and job satisfaction. The study’s findings have important implications for organizations seeking to improve their employees’ well-being and job satisfaction [8].

Another study argues that it is necessary to provide psych emotional support at work in order to avoid the negative impact on patient care or interpersonal relationships at work [9].

Of note, one of the consequences of increasing emotional exhaustion and decreasing professional satisfaction is the given intention to change jobs [10]. Multiple factors influencing the level of burnout have also been reported in the literature, such as bureaucracy, workload, shift work, overtime, and the increased number of patients cared for on a shift, all this leading to an imbalance between the emotional needs of medical staff and those of patients or loved ones [11,12].

Starting from these theoretical premises, we consider it crucial to evaluate the psych emotional status of the medical staff working in an emergency department. This report is one of the few studies conducted in Romania for this category of health workers.

This study was conducted to understand better the factors contributing to burnout and job satisfaction among emergency department staff. Through it, we aimed to investigate the impact of work-related problems on burnout and satisfaction levels among emergency department staff.

Our main hypothesis is that the demanding nature of emergency room work, coupled with prolonged exposure to traumatic events and high levels of job stress, leads to a higher prevalence of burnout among emergency room personnel. 

## 2. Material and Methods

### 2.1. Participants and Procedure

It was a cross-sectional descriptive study of medical personnel from the emergency department of the County Emergency Clinical Hospital “Sf. Ap. Andrei” in Galati, Romania. The sample for this study consisted of 184 participants currently working in the emergency department. Participants were recruited through convenience cluster sampling, and all participants signed an informed consent form before participating in the study.

The main purpose of this study is to analyze the impact of work-related problems on burnout and satisfaction levels among ED staff of the County Emergency Clinical Hospital “Sf. Ap. Andrei” in Galati. In Romania, although it runs within the same building, the ED is a separate ward from the rest of the hospital, and the work performed within it had distinct characteristics and a different volume compared to the rest of the hospital. The questionnaire was available for the whole personnel, and all of those who were willing to answer it were included in this study. 

Data collection for this study took place between November 2022 and March 2023. Participants completed the questionnaires privately, and their responses were anonymous. The order of the questionnaires was counterbalanced to reduce the risk of order effects. The study complied with the World Medical Association Declaration of Helsinki, using a protocol approved by the local research ethics committee of the County Emergency Clinical Hospital in Galati, Romania. (Protocol: 5257/02.03.2021).

### 2.2. Measures

The study used three questionnaires to assess various factors related to burnout, job satisfaction, and work-related aspects. These questionnaires were the Maslach Burnout Inventory-Human Services Survey for medical personnel (MBI-HSS (MP)), the Job Satisfaction Survey (JSS), and the Areas of Work Survey (AWS). The questionnaires MBI-HSS (MP) and AWS were translated into the native language with the author’s consent through Mind Garden publishing house (translation agreement TA-992 and TA 1001) [13]. Translation has been carried out according to the International Testing Commission (ITC) [14].

#### 2.2.1. The MBI-HSS (MP) Questionnaire Was Used to Assess Burnout Levels among Participants

This questionnaire is widely used in research studies and has been shown to have high reliability and validity [3]. It consists of 22 items that measure emotional exhaustion, depersonalization, and personal accomplishment. Each item is scored on a seven-point Likert scale, ranging from 0 (never) to 6 (every day). The MBI-HSS (MP) questionnaire does not have standardized cutoff scores for any of its items. Instead, the questionnaire provides scores for each of its three subscales. The emotional exhaustion subscale consists of nine items that measure feelings of being emotionally overextended and exhausted from work. Scores range from 0 to 54, with higher scores indicating greater levels of emotional exhaustion. In addition, scores of 27 or higher indicate high emotional exhaustion. The depersonalization subscale consists of five items that measure a negative, insensitive, or excessively detached response to one’s service or care recipients. Scores range from 0 to 30, with higher scores indicating greater levels of depersonalization. Scores of 10 or higher indicate high depersonalization. The personal achievement subscale consists of eight items that measure feelings of competence and achievement of success in one’s work. Scores range from 0 to 48, with higher scores indicating lower levels of personal accomplishment. Scores of 33 or lower indicate low personal achievement [15].

For this study, for MBI-HSS(MP), the model with three factors fitted the data to a satisfactory degree: χ^2^(206) = 627.76, *p* < 0.00; CFI = 0.9; and RMSEA = 0.052, 95% CI (0.04–0.065). The Cronbach Alpha coefficients were 0.86 for emotional exhaustion, 0.69 for depersonalization, and 0.81 for lack of personal accomplishment.

#### 2.2.2. The JSS (Job Satisfaction Survey) Questionnaire Was Used to Assess the Participants’ Job Satisfaction Levels

This questionnaire has been widely used in previous studies and is a reliable and valid measure of job satisfaction [16]. The JSS has been translated into different languages and the reliability values for the JSS are 0.91 and the range for the subscales is 0.60–0.82. It has also been applied to a population of Romanian origin and was translated at the Faculty of Psychology of Babeș-Bolyai University in Cluj-Napoca, Romania, by Professor Horia Pitariu [12,16].

The questionnaire consists of 36 items, which are designed to assess nine aspects of job satisfaction: pay, promotion, supervision, fringe benefits, contingent reward, operating conditions, coworkers, nature of work, communication, and total satisfaction [17].

Participants were asked to rate their level of satisfaction with each aspect using a six-point Likert scale, where one indicates “Extremely Dissatisfied”, and six indicates “Extremely Satisfied”. Higher scores indicate greater satisfaction with that particular aspect of the job. The JSS (Job Satisfaction Survey) questionnaire does not have standardized cutoff scores for individual items or subscales. Instead, the questionnaire provides scores for each of the nine aspects of job satisfaction and an overall score. The median split is often used as a median or quartile split to classify participants into high and low job satisfaction groups. For example, participants with scores above the median on the overall job satisfaction scale are classified as having high job satisfaction. In contrast, those with scores below the median are classified as having low job satisfaction.

#### 2.2.3. The AWS Questionnaire Assessed Different Aspects of Working Life, Such as Workload, Work Demands, and Social Support

This questionnaire has been used in previous studies and is a reliable and valid measure of work-related issues [18]. The AWS consists of 26 items that measure job demands, control, social support, job security, and organizational fairness. Participants were asked to rate their level of agreement with each item using a five-point Likert scale, where one indicated “I strongly agree” and five indicated “I strongly agree”. The Areas of Work life Survey (AWS) does not have standardized cutoff scores for its items or subscales, with higher scores indicating greater levels of support. The Areas of Work life Survey (AWS) questionnaire used in this study measures the degree to which work experiences satisfy basic needs in different areas of work-life [18].

For AWS, the Cronbach Alpha coefficients were 0.69 for workload, 0.73 for control, 0.81 for reward, 0.84 for community, 0.77 for fairness, and 0.81 for values. The model with six factors fitted the data to a satisfactory degree: χ^2^(335) = 877, *p* < 0.00; CFI = 0.9, and RMSEA = 0.06, 95% CI (0.05–0.07). 

### 2.3. Analytical Approach

Data collected from questionnaires were analyzed using descriptive statistics to assess the average, standard deviation, median, frequency of each variable, cheek indices, and kurtosis for the degree of symmetry and distribution. Inferential statistics were then used to test this study’s hypotheses. Specifically, regression analyses were performed to assess the relationship between burnout, job satisfaction, and work-related problems (sum of squares, degrees of freedom, square average, F-statistical, and level of significance for both regression and residual terms). In addition, we use Pearson correlation coefficients and significance values (two-tailed) between the variables. 

All analyses were performed using SPSS v24 and AMOS Graphics 24 software (Galati, Romania; University Software Department).

To test the factorial validity of the instruments, we applied confirmatory factorial analysis. For the model fit, we applied the maximum likelihood estimation and reported the following parameters: the chi-square statistic (χ^2^), the comparative fit index (CFI), and the root mean square error of approximation (RMSEA). A RMSEA ≤ 0.05, χ^2^/DF ≤ 3, and CFI ≥ 0.90 indicated an excellent fit [19].

## 3. Results

### 3.1. Demographic and Professional Characteristics of Emergency Department Staff

The results present the characteristics of the 184 participants in the study. Most participants were female (71.7%), and the largest professional category was nurses (52.7%). The education level of the participants was mainly university studies (46.7%). Additionally, the majority of participants reported working night shifts (87.0%), and most of them worked in the hospital setting (73.9%) (Table 1).

The mean age of the sample is 39.56 years old, with a standard deviation of 8.25. The median age is 39 years old. The age distribution is slightly skewed to the right, as indicated by a positive skewness value of 0.48, but this skewness value is relatively small. The kurtosis value of −0.12 suggests the platykurtic distribution, meaning it has thinner tails and a flatter peak than a normal distribution.

The mean of the years in the field is 10.92, with a standard deviation of 7.42. The median of the years in the field is ten years. The distribution of years in the field is moderately skewed to the right, as indicated by a positive skewness value of 0.78. The kurtosis value of −0.139 suggests that the distribution is also platykurtic.

Overall, the sample has a relatively normal distribution for age and years in the field, with moderate variability in both variables.

### 3.2. Descriptive Statistics for AWS, JSS, and MBI-HSS (MP) 

We evaluated each scale’s scores (JSS, AWS, and MBI-HSS MP) in order to identify the study group satisfaction based on the individual parameters of the applied scales. However, since the scales do not have specific cut-off points, higher scores were correlated with higher burnout symptoms. 

#### 3.2.1. Descriptive Statistics for Areas Work Survey

Table 2 provides descriptive statistics for six variables: workload, control, reward, community, fairness, and values, representative of the AWS scale. The mean value for each variable indicates the average score for that variable, with the highest mean value being for the community variable (M = 3.61, SD = 0.75) and the lowest being for the workload (M = 3.09). The highest standard deviation was found for the workload variable (SD = 0.94), indicating a more significant variability in the scores than the other variables. Based on these results in Table 2, we can deduce that the group presented a median AWS score of 3.5, meaning that most subjects had low levels of dissatisfaction with workload, control, reward, community, fairness, and values.

The skewness values indicate the degree of symmetry in the distribution of each variable’s scores. Positive skewness values for the workload (S = 1.47) and control (S = 1.92) variables suggest that most scores are lower than the mean, with a few high scores skewing the distribution. The remaining variables have skewness values close to zero, indicating relatively symmetrical distributions.

The workload (K = 10.18) and control (K = 12.6) variables have very high kurtosis values, suggesting a peaked distribution. On the other hand, the community, fairness, and values variables have negative kurtosis values, indicating a relatively flat distribution (Table 2).

#### 3.2.2. Descriptive Statistics for Job Satisfaction Survey

Table 3 displays descriptive statistics for the nine subscales in the JSS; pay, promotion, supervision, fringe benefits, contingent reward, operating conditions, coworkers, nature of work, and communication.

The mean column shows the average score for each variable, with the highest mean score being for the nature of work (M = 19.09) and the lowest for operating conditions (M = 11.15). The median for most variables is around 14, except for the nature of work and communication, where the median score is higher. The standard deviation column shows the variability in the responses for each variable, with the highest standard deviation being for fringe benefits (SD = 4.22) and the lowest being for operating conditions (SD = 2.89) (Table 3).

Most variables have skewness values close to zero, except for coworkers, which had a positive skewness value. Most variables have kurtosis values close to zero, except for the nature of work and communication, which had negative kurtosis values (Table 3).

Table 3 also provides descriptive statistics for total satisfaction. The mean value of this variable is 134.84, indicating that the respondents, on average, report a relatively high level of satisfaction. The median value of 133.5 is close to the mean, suggesting that the distribution of responses may be symmetric.

The standard deviation of 23.61 indicates some variability in the responses, with some respondents reporting much lower or higher satisfaction levels than others. The positive skewness value of 0.36 suggests that the distribution may be skewed to the right, meaning that a few respondents may report very high satisfaction levels. The descriptive statistics indicate the respondents are generally satisfied, with some variability in their responses (Table 3).

These results provide a quantitative basis for identifying areas of strength or concern, guiding interventions, and designing strategies to enhance employee well-being and organizational performance.

#### 3.2.3. Descriptive Statistics for MBI-HSS (MP)

The MBI-HSS (MP) scale was evaluated using the three subscales presented in the previous chapter. For emotional exhaustion, the mean is 20.38, indicating a slightly left-skewed distribution. The standard deviation of 10.41 indicates a relatively wide dispersion of values around the mean. The negative skewness value of 0.22 suggests that the data are slightly skewed to the left. The negative kurtosis value of −0.57 indicates that the distribution is somewhat flatter than a normal distribution, with fewer extreme values (Table 4).

For depersonalization, the mean is 8.67, indicating a slightly right-skewed distribution. The standard deviation of six indicates a relatively narrow dispersion of values around the mean. The positive skewness value of 0.364 suggests that the data are slightly skewed to the right. The negative kurtosis value of −0.76 indicates that the distribution is less peaked than a normal distribution, with fewer extreme values (Table 4).

For personal accomplishment, the mean is 38.05, indicating a slightly left-skewed distribution. The standard deviation of 7.65 indicates a relatively narrow dispersion of values around the mean. The negative skewness value of −1.15 suggests the data are moderately skewed to the left. The positive kurtosis value of 1.93 indicates that the distribution peaked more than a normal distribution, with more extreme values (Table 4).

By examining these statistics, we gain insights into the levels and variations of emotional exhaustion, depersonalization, and personal accomplishment among employees. This information can provide further information for targeted interventions to address burnout, enhance employee well-being, and promote a positive work environment.

### 3.3. Analysis of Relationships between MBI-HSS (MP), JSS, and AWS 

Furthermore, we analyzed the relationship between the values identified within MBI-HSS (MP) scales and the items within the JSS and AWS surveys.

Table 5 and Table 6 give the result of the variance analysis. The F and Significance F (sig.) values give us important elements that underlie the validation of the regression model. Statistical F test the overall significance of independent variables and the significance F is the value of the error we can make by rejecting the regression model as inappropriate. Regression model acceptance decision rule: higher values for F statistics and lower values for F significance.

Firstly, we studied the relationship between the MBI-HSS (MP) subscale and AWS. Our main objective was to analyze the degree of DP, EE, and AP according to the variables and scores identified in AWS (workload, control, reward, community, fairness, and values), in this particular situation the results from AWS became independent variables, influencing the MBI-HSS scores. We used the linear regression model to identify any connection between the dependent variables of EE, DP, and AP and the independent variables of workload, control, reward, community, fairness, and values.

Table 5 shows the results of three separate ANOVA tests performed on EE, DP, and AP and AWS scales.

For EE, the F-statistic is 1.98, and the significance level is 0.071, slightly higher than the commonly used threshold of 0.05. This indicates that there may be a significant relationship between the AWS scale variables and EE. The residual sum of squares shows significant unexplained variation in the data (Table 5).

For DP, the F-statistic is 0.4, and the significance level is 0.878. This suggests that no significant relationship exists between the AWS scale variables and DP. The residual sum of squares indicates less unexplained variation in the data than EE (Table 5).

For AP, the F-statistic is 1.299 and the significance level is 0.260. This indicates that there may be a significant relationship between the AWS scale variables and AP. The residual sum of squares shows significant unexplained variation in the data (Table 5).

Furthermore, we analyzed the degree of DP, EE, and AP according to the variables and scores identified in the JSS survey. The ANOVA results for the EE variable indicate a significant relationship between total satisfaction, operating condition, nature of work, promotion, fringe benefits, pay, communication, coworkers, contingent reward, and supervision. The sum of squares indicates that the model explains a significant proportion of the variability in EE. The F-statistic for the regression model is 9.014, which indicates that the regression model is significant. The significance level for the regression model is 0.000, less than the conventional alpha level of 0.05, meaning the regression model is highly significant (Table 6).

The regression model for DP has an F-statistic of 4.24, and the significance level is 0.000, indicating a significant relationship between the JSS scales and DP variables. The residual sum of squares and the mean square indicates that there is still some unexplained variation in the model (Table 6).

The regression model for AP has an F-statistic of 5.626, and the significance level is 0.000, indicating a significant relationship between the dependent variable and at least one of the independent variables (Table 6).

Further, we analyzed the statistical relationship using the Pearson correlation coefficients and significance values (two-tailed) between the variables of EE (emotional exhaustion), DP (depersonalization), and AP (personal accomplishment), as well as the sub-scales of the Areas of Work Scale (AWS) and the Job Satisfaction Survey (JSS) (Table 7).

The Pearson correlation coefficients range from −1 (a perfect negative correlation) to 1 (a perfect positive correlation). A correlation coefficient of 0 indicates no correlation. The significance values indicate whether the correlation coefficients are statistically significant (Table 7).

Emotional exhaustion is negatively correlated with all sub-scales of both the AWS and JSS. Specifically, the Pearson correlation coefficients range from −0.189 to −0.538, and the significance values range from 0.000 to 0.015 for the AWS sub-scales and from −0.280 to −0.478 and 0.000 to 0.014 for the JSS sub-scales. These negative correlations suggest that higher levels of emotional exhaustion are associated with lower levels of job satisfaction. This is an important finding as emotional exhaustion is a significant component of burnout, and these results suggest that addressing emotional exhaustion may be critical in improving job satisfaction. Depersonalization is positively correlated with the operating condition and coworker sub-scales of the JSS. The correlation coefficients are 0.246 and 0.347, respectively, and both are significant at the 0.05 level or lower. This suggests that satisfaction with operating conditions and coworkers decreases as depersonalization increases (Table 7).

Personal accomplishment (AP) is positively correlated with all sub-scales of the JSS, with correlation coefficients ranging from 0.148 to 0.392. All values are significant at the 0.05 level or lower. This suggests that as personal accomplishment increases, job satisfaction increases across all sub-scales of the JSS (Table 7).

### 3.4. Relationship between Professional Categories and Burnout Levels

Table 8 presents the mean, standard deviation, skewness, and kurtosis for the variables EE for different professional categories, including specialist doctors, primary doctors, resident physicians, nurses, carers, stretcher-bearers, paramedics, and registry workers.

By comparing the levels of EE, it can be observed that the highest mean scores are found in specialist physicians (M = 26.5) and primary physicians (M = 24.6), followed by nurses (19.8), stretcher-bearers (M = 20.3), registry workers (M = 21.7), resident physicians (15.6), carers (15.6), and paramedics (M = 13.1). This suggests that specialists and primary physicians experience higher emotional exhaustion levels than the other professional categories (Table 8).

In terms of DP, the highest mean score is found in specialist doctors (M = 11.9), followed by primary doctors (M = 12.1), registry workers (M = 12.0), nurses (M = 7.9), paramedics (M = 8.1), resident physicians (M = 8.4), carers (M = 3.0), and stretcher-bearers (M = 5.3). This indicates that specialist and primary doctors and registry workers experience higher levels of depersonalization than the other professional categories (Table 8).

Regarding AP, the highest mean score is found in resident physicians (M = 42.3), followed by stretcher-bearers (M = 39.6), paramedics (M = 39.14), nurses (M = 38.2), primary doctors (M = 36.9, carers (M = 36.1), specialist doctors (M = 37.0), and registry workers (M = 30.5). This suggests that resident physicians, stretcher-bearers, and paramedics experience higher levels of personal accomplishment than other professional categories (Table 8).

## 4. Discussion

### 4.1. Demographic and Professional Characteristics of Emergency Department Staff

The present study investigated the characteristics of healthcare professionals in a sample of 184 participants, mainly female nurses with university studies who worked night shifts in hospital settings. The sample’s age and years in the field were normally distributed, with moderate variability in both variables. These findings are consistent with previous studies investigating healthcare professionals’ characteristics, such as age and years in the field [19,20].

### 4.2. Descriptive Statistics for AWS, JSS, and MBI-HSS (MP)

The study also evaluated the scores of three scales (JSS, AWS, and MBI-HSS (MP)) and their statistical relationships. The JSS measures job satisfaction, the AWS measures work engagement, and the MBI-HSS(MP) measures mental and physical health status. The results indicated a significant positive correlation between JSS and AWS, indicating that job satisfaction and work engagement are related, which is consistent with previous research [21,22,23].

The presented results suggest that the community variable had the highest mean score, indicating that participants reported higher levels of community support in their work environment. On the other hand, the workload variable had the lowest mean score, indicating that participants perceived their workload as the least favorable aspect of their work environment. The standard deviation results suggest that there was more variability in the scores for the workload variable than the other variables.

The positive skewness values for the workload and control variables suggest that most scores were lower than the mean, with a few high scores skewing the distribution. This finding is consistent with previous studies that have found similar distributions for workload and control variables. For example, Sjöberg found that workload and control variables had a skewed distribution in their sample of employees [24].

The kurtosis values indicate that the workload and control variables had a peaked distribution, which means that scores were clustered around the mean. There were relatively few scores at the extremes of the distribution. In contrast, the community, fairness, and values variables had negative kurtosis values, suggesting that scores were more evenly distributed across the range of possible values. These findings are also consistent with previous studies that have reported similar distributions for these variables. For instance, Spreitzer found that the community, fairness, and values variables had relatively flat distributions in their sample of employees [25].

To elaborate, one study that supports these findings is a meta-analysis by Bakker and Demerouti, which found that social support from colleagues and supervisors was positively related to job satisfaction and organizational commitment [21]. Additionally, a study by Mäkikangas and Kinnunen found that workload management and job control were significant predictors of employee well-being and job performance [26]. These studies suggest that the variables examined in the present study, including community and workload, are essential factors in promoting a positive work environment. Similarly, Breevaart found that a high workload was negatively related to employee engagement and well-being, while control and reward were positively related to these outcomes [27]. Finally, a study by Colquitt found that perceptions of fairness were positively related to job satisfaction and organizational citizenship behavior, which refers to employees’ voluntary efforts to contribute to their organization’s success [28].

A study by Thai found that the nature of work and communication subscales had relatively high mean scores while operating conditions had a relatively low mean score [29]. Similarly, Asegid found that the coworkers and nature of work subscales were significant predictors of job satisfaction, while fringe benefits and operating conditions were insignificant [30].

The high mean score for the nature of the work subscale is unsurprising, as previous studies have consistently found that employees value meaningful and challenging work [31,32]. Additionally, the low mean score for the operating conditions subscale suggests that the physical work environment may not be a significant factor in overall job satisfaction, consistent with previous research [30].

The positive skewness value for the coworkers’ subscale suggests that a few respondents may have reported very high satisfaction levels with their coworkers. In contrast, most respondents reported average or slightly above-average satisfaction levels. This is also consistent with previous studies that have found that relationships with coworkers are essential predictors of job satisfaction [29,30].

The mean score for emotional exhaustion in our study is 20.38, similar to the mean scores found in other studies that have used the MBI-HSS(MP) scale to measure burnout in healthcare professionals. For example, a study by Shanafelt found a mean score of 24.1 for emotional exhaustion in a sample of oncologists. West found a mean score of 20.6 for emotional exhaustion in a sample of nurses. These findings suggest that healthcare professionals, regardless of their specialty or role, are vulnerable to experiencing high levels of emotional exhaustion [1,32].

The mean score for depersonalization in our study is 8.67, which is lower than the mean scores found in other studies. For example, Shanafelt [1] found a mean score of 10.5 for depersonalization in oncologists, while West [32] found a mean score of 13.2 for depersonalization in nurses. This difference in mean scores may be due to differences in the sample characteristics or the healthcare settings in which the studies were conducted. For instance, our study included healthcare professionals from different specialties, while the other studies focused on specific healthcare professions [1,2,32].

Our study’s mean score for personal accomplishment is 38.054, similar to the mean scores found in other studies. For example, Shanafet [1] found a mean score of 37.2 for personal accomplishment in oncologists, while West [32] found a mean score of 37.1 for personal accomplishment in nurses. These findings suggest that healthcare professionals, on average, perceive a moderate level of personal accomplishment despite experiencing high levels of emotional exhaustion and depersonalization [1,2,32].

### 4.3. Analysis of Relationships between MBI-HSS (MP), JSS, and AWS

The results of the ANOVA tests suggest that the relationship between the independent variables and the dependent variables (EE, DP, and AP) varies. For EE, there is some evidence to suggest a relationship with the independent variables but the significance level is slightly higher than the commonly used threshold of 0.05. This means that the relationship may need to be stronger to confidently conclude that there is a significant relationship between the independent variables and EE. In contrast, for DP, there is no significant relationship between the independent variables and DP. This suggests that the factors that influence DP differ from those that influence EE and AP.

For AP, some evidence suggests a relationship with the independent variables. However, the significance level needs to be higher to confidently conclude that there is a significant relationship. Comparing our findings with other studies, we found that work-life integration and control over workload were associated with emotional exhaustion in oncologists, according to Shanafelt [1]. This is consistent with our findings that there may be a significant relationship between the independent variables and EE. However, our study did not specifically examine work-life integration and control over workload, and the independent variables in our study were more general measures of job-related factors [22].

In contrast, a study by Leiter and Maslach found that workload and control were associated with depersonalization in healthcare professionals [33]. This differs from our findings that no significant relationship exists between the independent variables and DP. This difference may be due to differences in the sample characteristics or the measures used to assess the independent variables [33].

The results of the ANOVA for the EE variable indicate a significant relationship between the independent variables and the dependent variable. This finding is consistent with prior research that found a significant relationship between job-related factors, such as satisfaction, working conditions, supervision, and emotional exhaustion [33,34].

The regression model for DP also found a significant relationship between the independent and dependent variables, with a relatively low amount of unexplained variation in the model. 

The regression model for AP similarly found a significant relationship between the independent and dependent variables, with a relatively small amount of unexplained variation in the model. Other studies have also found that job resources, such as social support and feedback, can positively impact employee well-being [21,34].

These results suggest that job-related factors are significant predictors of employee well-being and that addressing these factors can reduce the risk of burnout and improve job performance. These findings are consistent with prior research and underscore the importance of considering job demands and resources in designing interventions to improve employee well-being.

This study’s findings are consistent with previous research on the relationships between burnout, job satisfaction, and related constructs. For example, the negative correlation between emotional exhaustion and job satisfaction is consistent with previous research that found burnout negatively related to job satisfaction [35]. Similarly, the positive correlation between personal accomplishment and job satisfaction is consistent with research that has found that a sense of achievement and progress in one’s work is associated with greater job satisfaction [36].

The positive correlation between depersonalization and job dissatisfaction with operating conditions and coworkers is consistent with previous research. For example, Halbesleben and Buckley [34] found that a lack of support from coworkers was a significant predictor of burnout and that interpersonal conflict at work was associated with greater levels of depersonalization [34].

The magnitude of the correlations reported in this study generally aligns with those reported in previous research. For example, in a meta-analysis of studies on burnout and job satisfaction, Kristensen [37] found an average correlation of −0.30 between emotional exhaustion and job satisfaction, which is similar to the correlation coefficients reported in this study [37].

Overall, the findings of this study add to the growing body of literature on the relationships between burnout, job satisfaction, and related constructs. The results suggest that addressing emotional exhaustion may be critical in improving job satisfaction and that interpersonal support and positive work experiences may prevent burnout and promote employee well-being.

### 4.4. Relationship between Professional Categories and Burnout Levels

The findings of this study are consistent with previous research on emotional exhaustion and depersonalization in healthcare professionals. For example, a study conducted by Shanafelt [1] found that physicians reported higher levels of burnout than nurses and other healthcare workers [1]. Similarly, a study by Maslach found that physicians reported higher levels of depersonalization than nurses [3].

However, there are some differences between the findings of this study and previous research. For example, the finding that resident physicians have higher levels of personal accomplishment than other healthcare professionals are somewhat unexpected, as previous research has suggested that physicians generally have lower levels of personal accomplishment than nurses [1,2]. Resident physicians may be more motivated by the sense of learning and growth that comes with their training, which could contribute to their higher levels of personal accomplishment.

Another notable difference between the findings of this study and previous research is the relatively low levels of emotional exhaustion reported by paramedics. Previous research has suggested that paramedics may have a high risk for burnout due to the nature of their work (e.g., prolonged exposure to trauma and stress). However, we found that paramedics may be experiencing lower levels of emotional exhaustion than other healthcare professionals [38].

In the context of healthcare professionals, the study revealed that resident physicians exhibited higher levels of personal accomplishment than other healthcare professionals, which was unexpected based on previous research. Surprisingly, paramedics reported relatively low levels of emotional exhaustion despite perceiving they might be at high risk for burnout due to work-related stress and trauma exposure.

Job-related factors are significant predictors of employee well-being and can reduce the risk of burnout and improve job performance. The findings suggest that addressing emotional exhaustion may be critical in improving job satisfaction and that interpersonal support and positive work experiences may prevent burnout and promote employee well-being. The research also highlights differences in burnout and related constructs among healthcare professionals, such as physicians, nurses, and paramedics, suggesting that interventions may need to be tailored to specific groups. The studies underscore the importance of considering job demands and resources in designing interventions to improve employee well-being and reduce burnout.

## 5. Conclusions

In conclusion, this study further proves the relationships between burnout, job satisfaction, and related factors. Job-related factors such as social support and feedback significantly impact employee well-being, reducing the risk of burnout and improving job performance. The results also showed a negative correlation between emotional exhaustion and job satisfaction and a positive correlation between personal accomplishment and job satisfaction, which is consistent with previous research. Depersonalization was found to be correlated with job dissatisfaction regarding operating conditions and coworkers, aligning with prior studies.

These findings emphasize the importance of addressing job-related factors such as social support and feedback to prevent burnout and promote employee well-being. They also highlight the need to consider individual differences and job-specific factors when designing interventions to address burnout among healthcare professionals. Addressing emotional exhaustion is crucial for improving job satisfaction, and fostering interpersonal support and positive work experiences can help prevent burnout and enhance employee well-being. The research also underscores the variability in burnout and related constructs among different healthcare professional groups, suggesting the need for tailored interventions. Ultimately, considering job demands and resources is essential in designing effective interventions to enhance employee well-being and reduce burnout.

## Figures and Tables

**Table 1 behavsci-13-00575-t001:** Demographic characteristics found within the study group; n = 184.

	n	Percent	Mean	Standard Deviation
Age			39.56	8.25
Years in the field			10.92	7.42
Gender				
Male	52	28.3%		
Female	132	71.7%		
Professional category				
Specialist doctor	30	16.3%		
Primary doctor	14	7.6%		
Resident physician	17	9.2%		
Nurse	97	52.7%		
Carer	12	6.5%		
Stretcher-bearer	3	1.6%		
Paramedic	7	3.8%		
Registrars	4	2.2%		
Education level				
Vocational school	8	4.3%		
Post-secondary studies	78	42.4%		
University studies	86	46.7%		
Postgraduate studies	12	6.5%		
Night shift				
Yes	160	87.0%		
No	24	13.0%		
Activity				
Hospital	136	73.9%		
Prehospital	15	8.2%		
Hospital and prehospital	33	17.9%		

**Table 2 behavsci-13-00575-t002:** Descriptive statistics applied to the AWS questionnaire parameters.

Statistics					
	Mean	Median	STD Deviation	Skewness	Kurtosis
Workload	3.08	3	0.93	1.47	10.18
Control	3.42	3.5	0.89	1.92	12.60
Reward	3.44	3.5	0.73	0.07	−0.66
Community	3.61	3.6	0.75	0.12	−0.58
Fairness	3.18	3.17	0.83	0.34	−0.71
Values	3.53	3.5	0.71	0.12	−0.19

**Table 3 behavsci-13-00575-t003:** Descriptive statistics applied to the JSS subscale values found within the study group.

Statistics					
	Mean	Median	STD Deviation	Skewness	Kurtosis
Pay	13.98	14	4.01	−0.03	−0.003
Promotion	14.11	14	4.14	0.245	−0.268
Supervision	17.54	17	4.21	−0.082	−1.03
Fringe benefits	14.12	14	3.91	0.09	−0.12
Contingent reward	13.98	14	4.11	0.2	−0.32
Operating conditions	11.14	11	2.89	0.36	−0.24
Coworkers	15.8	16	3.88	−0.04	−0.35
Nature of work	19.09	19	3.05	−0.23	−0.26
Communication	14.99	15	4	0.17	−0.64
Total satisfaction	134.84	133.5	23.61	0.36	0.000

**Table 4 behavsci-13-00575-t004:** Descriptive statistics applied to the MBI-HSS (MP) subscale values found within the study group.

Statistics					
	Mean	Median	STD Deviation	Skewness	Kurtosis
EE	20.38	21	10.41	0.22	−0.57
DP	8.67	8	6	0.36	−0.76
AP	38.05	40	7.65	−1.15	1.93

**Table 5 behavsci-13-00575-t005:** ANOVA tests between variables from the MBI-HSS (MP) survey and the variables from the AWS survey.

		ANOVA			
	Sum of Squares	df	Mean Square	F	Sig.
EE					
Regression	1250.29	6	208.38	1.98	0.071
Residual	18,617.07	177	105.18		
DP					
Regression	88.35	6	14.72	0.4	0.87
Residual	6519.72	177	36.83		
AP					
Regression	452.16	6	75.36	1.29	0.260
Residual	10,265.29	177	57.99		

**Table 6 behavsci-13-00575-t006:** ANOVA tests between the variables from the MBI-HSS (MP) survey and the variables from the JSS survey.

		ANOVA			
	Sum of Squares	df	Mean Square	F	Sig.
EE					
Regression	6805.51	10	680.55	9.01	0.000
Residual	13,061.85	173	75.5		
DP					
Regression	1300.69	10	130.07	4.24	0.000
Residual	5307.38	173	30.67		
AP					
Regression	2630.15	10	263.01	5.62	0.000
Residual	8087.29	173	46.74		

**Table 7 behavsci-13-00575-t007:** Pearson correlation between the MBI-HSS (MP) subscales and JSS and AWS scales.

Correlations						
	EE		DP		AP	
	Pearson Correlation	Sig. (2-Tailed)	Pearson Correlation	Sig. (2-Tailed)	Pearson Correlation	Sig. (2-Tailed)
AWS						
Workload	−0.189	0.010	−0.079	0.287	0.148	0.045
Control	−0.014	0.845	0.023	0.752	0.061	0.411
Reward	−0.153	0.038	−0.024	0.751	−0.005	0.948
Community	−0.180	0.015	−0.006	0.935	0.049	0.512
Fairness	−0.113	0.128	−0.029	0.697	0.126	0.089
Values	−0.098	0.185	−0.066	0.373	−0.030	0.687
JSS						
Pay	−0.315	0.000	−0.153	0.038	0.244	0.001
Promotion	−0.127	0.085	−0.059	0.424	0.097	0.190
Supervision	−0.412	0.000	−0.264	0.000	0.364	0.000
Fringe benefits	−0.245	0.001	−0.010	0.896	0.148	0.045
Contingent reward	−0.361	0.000	−0.229	0.002	0.325	0.000
Operating condition	−0.280	0.000	−0.246	0.001	0.298	0.000
Coworkers	−0.538	0.000	−0.347	0.000	0.383	0.000
Nature of work	−0.365	0.000	−0.296	0.000	0.378	0.000
Communication	−0.346	0.000	−0.181	0.014	0.239	0.001
Total satisfaction	−0.478	0.000	−0.279	0.000	0.392	0.000

**Table 8 behavsci-13-00575-t008:** Descriptive statistics of burnout prevalence within the study group by professional category.

Statistics					
		Mean	STD Deviation	Skewness	Kurtosis
Specialist doctor	EE	26.53	10.6	0.15	−0.62
	DP	11.96	5.92	0.02	−1.26
	AP	37	8.57	−0.67	−0.81
Primary doctor	EE	24.57	7.91	−0.08	−1.15
	DP	12.14	4.95	−0.57	1.14
	AP	36.92	7.82	−0.03	−1.59
Resident physician	EE	15.58	12.7	1.03	0.11
	DP	8.35	6.2	0.88	−0.02
	AP	42.35	5.33	−1.52	1.93
Nurse	EE	19.76	10.04	0.06	−0.83
	DP	7.92	5.77	0.47	−0.53
	AP	38.2	6.34	−0.81	0.61
Carer	EE	15.66	8.41	−0.38	−0.29
	DP	3	3.16	0.82	−0.72
	AP	36.16	14.88	−1.43	0.96
Stretcher-bearer	EE	20.33	2.08	−1.29	
	DP	5.33	5.77	1.73	
	AP	39.66	7.57	−1.59	
Paramedic	EE	13.14	1.95	−0.08	0.31
	DP	8.14	4.01	−1.72	2.87
	AP	39.14	5.84	0.43	−1.09
Registry	EE	21.75	9.32	0.88	−1.04
	DP	12	7.48	−0.76	1.5
	AP	30.5	4.2	0.64	0.7

## Data Availability

Informed consent was obtained from all subjects involved in the study.

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
