# Peer review of "The Impact of Work-Related Problems on Burnout Syndrome and Job Satisfaction Levels among Emergency Department Staff"

_behavsci, 2023, doi:10.3390/bs13070575_

Round 1
Reviewer 1 Report
Dear Author
Thank you for the opportunity to read your research.
Here are some recommendations to improve your work.
1. The use of validated tools to identify burnout, job satisfaction, is accurate. However, I would like to know if these instruments were provided in their English version or if they were translated into the native language of the sample studied. If translated into the language of the sample, was a cross-culturally adapted tool used? If the original version was used, was the sample sufficiently familiar with the language of the instrument? Could the instrument be understood by the sample members?
2. Indicates that a convenience sampling of health professionals was conducted. Was cluster sampling or proportionate separation of the sampled professionals performed? This aspect may affect the representativeness of the categories and the results obtained.
3. Improve the editing of Table 1. It is unclear.
4. Synthesize the wording of the conclusions.... They are excessively long.
5. Review the evidence that supports both the background and the discussion of the article; the number of references older than 5 years, and 10 years, is excessive. Replace with more current evidence.
Reviewer 2 Report
This article provided insight about the factors that lead to burnout among emergency department staff. Overall this topic is important and valuable, which help us understand the burnout among this specific group. At the same time, I have some suggestions for the improvement of this article. Introduction section: I suggested authors to provide more previous studies on the burnout among emergency department staff or relevant samples. Comparing with very lengthy results and discussion, the introduction section is very short. Furthermore, authors could point out how the current work made contribution to the field. Do authors have any hypotheses? Even this is an exploratory study, I think there should be some hypotheses, which help to further clarify what this study aim to solve. Later sections mentioned "hypothesis" but I cannot see what are the core hypotheses at Introduction section. Methods section: I don't know whether it is necessary to provide so much detailed discussion about scales. Typically authors could briefly note which scale they used and reported the psychometric information and that should be enough. Please ingore this if this is the requirement of the journal. Results I don't quite understand ANOVA (Table 5). The authors said the independent variables are AWS survey? That survey have six dimensions and each dimension has a score (Table 2). Thus, how this survey could be used as independent varaibles? Please clarify if authors insist to use this statistics, currently this part is too confusing. Furthermore, based on the significant F values, authors should further report comparison between levels of independent variable. Additionally, in fact, I think a correlations between MBI-HSS and AWS seems enough to explore the relationship between two scales. If I misunderstand something, please correct me.Mean and SD don't need so much numbers in decimal places. Typically two numbers are enough (e.g., 3.0859 to 3.09). Too many focus on details like skewness values or kurtosis values are not very meaningful and biased from the major purpose of this article. I see authors briefly discuss these findings, that's good, but what is the theoretical or practical meaning of these findings? I mean, if this part is important, please mention it earlier. If they are not very meaningful, I think authors could delete them. By the way, authors did not mention this in Statistic Approach section. Overall, when I read from the beginning to the end of the manuscript, I feel the details and their purposes are confusing. Similarly, as noted above, I think the ANOVA could focus on F values, and then the comparisons between levels of the independent variable. Currently the detailed reports like "sum of squares of 1250.295, six degrees of freedom, and 266 a mean square of 208.383" are really redunant, which make readers hard to follow the core analysis of the article. I also want to suggest authors to use some sub-headings for Results section. Currently it is too long and hard to follow. Discussion section Overall, the discussion section is OK, but I still suggest subheadings to make it easier for readers. I also suggest authors to further highlight the contribution and meaning (practical implication) of their findings, or expand how this article "add to the growing body of literature". Currently, this work is simply a report of current status. It finds similarity and difference when compared to previous works, but the contributions (and inspiration for future) is not very impressive.
Round 2
Reviewer 2 Report
I appreciated for the efforts for revising and responding according to my previous comments. Overall the authors well response to the comments.
I still suggested authors to consider divide their discussion with subheadings, which will be helpful for readers to capture the major points.
The "variables" in headings of Table 5 and 6 are still confusings, if possible, please consider how to help readers better understanding them. Similarly, line 330, "the variables and scores identified in AWS", adding some explanation such as (i.e., XXXX, XXXX) might be good.
